# Safety and Efficacy of Prostaglandin Analogues in the Immediate Postoperative Period after Uneventful Phacoemulsification

**Eleftherios Anastasopoulos [1,\*], Spyridon Koronis [1], Artemis Matsou [2], Maria Dermenoudi [1,†], Nikolaos Ziakas [1] and Argyrios Tzamalis [1]**

[1] Department of Ophthalmology, General Hospital Papageorgiou, 56429 Thessaloniki, Greece; spyridonkoronis@gmail.com (S.K.)
[2] Corneoplastic Unit, Queen Victoria Hospital, East Grinstead RH19 3DZ, UK
[\*] Correspondence: eleftherios70@gmail.com; Tel.: +30-231332-3572
[†] Current Address: Health Center of Neapolis, 56727 Thessaloniki, Greece.

**Abstract:** Prostaglandin analogues (PGAs) have been associated with the development of pseudophakic macular edema (PME) in complicated cataract cases, but evidence on their effects in uncomplicated phacoemulsification remains controversial. This two-arm, prospective, randomised study included patients with glaucoma or ocular hypertension under PGA monotherapy who were scheduled for cataract surgery. The first group continued PGA use (PGA-on), while the second discontinued PGAs for the first postoperative month and reinitiated use afterwards (PGA-off). Topical non-steroidal anti-inflammatory drugs (NSAIDs) were routinely administered to all patients during the first postoperative month. The patients were followed up for three months and the primary outcome was PME development. Secondary outcomes were corrected distance visual acuity (CDVA), central and average macular thickness (CMT and AMT), and intraocular pressure (IOP). The analysis included 22 eyes in the PGA-on group and 33 eyes in the PGA-off group. No patient developed PME. CDVA was not significantly different between the two groups ($p = 0.83$). CMT and AMT showed a small but statistically significant increase until the end of follow-up ($p < 0.001$). Mean IOP values had no significant differences between the groups at each visit ($p > 0.05$). At the end of follow-up, the IOP values were significantly lower than baseline in both groups ($p < 0.001$). In conclusion, PGA administration with concomitant topical NSAIDs appears to be a safe practice in the early postoperative period of uncomplicated phacoemulsification.

**Keywords:** prostaglandin analogues; pseudophakic macular edema; Irvine–Gass syndrome; cataract surgery; phacoemulsification

## 1. Introduction

Cataracts and glaucoma remain the two leading causes of blindness in people aged over 50 years old, with a prevalence of 15.2 and 3.2, million cases respectively [1]. Since both are very common conditions of the worldwide elderly population, cataractsarea frequent comorbidity in glaucoma patients.

Cystoid macular edema (CME) manifests as variably reduced visual acuity and may appear biomicroscopically as macular thickening, sometimes with visible fluid accumulation into cystoid spaces in the outer plexiform and inner nuclear layers [2]. CME is usually the result of an inflammatory process. Common etiological factors include diabetes mellitus, uveitis, and retinal vein occlusion, but CME may also be associated with cataract surgery [3].

Pseudophakic CME (PME), also termed Irvine–Gass syndrome, may uncommonly lead to vision loss following phacoemulsification even in uncomplicated cases [4]. It is generally accepted that the pathogenesis of PME is related to postoperative inflammation. Pro-inflammatory mediators compromise the blood–aqueous barrier, which enables them

to diffuse to the vitreous and subsequently the retina. In turn, they facilitate the breakdown of the inner and outer blood–retina barrier, which leads to the accumulation of intraretinal fluid [5]. These inflammatory molecules include platelet-activating factor, complement factors, lysosomal enzymes, nitric oxide, endothelin, vascular endothelial growth factor, insulin-like growth factor-1, and prostaglandins [6]. In the absence of risk factors, the incidence of PME has been reported at 1.17%. Diabetes mellitus, epiretinal membrane, retinal vein occlusion, pars plana vitrectomy for retinal detachment, a history of uveitis, as well as posterior capsule rupture are the most prominent risk factors for the development of PME [4].

Diagnostic imaging techniques include fluorescein angiography (FFA), where PME manifests as late leakage, and optical coherence tomography (OCT), where the intraretinal fluid appears as cystoid spaces of low reflectivity. The use of newer imaging techniques has also been suggested for the diagnosis of PME. Fundus autofluorescence displays pigment displacement from the hypofluorescent cysts, while OCT angiography shows decreased capillary perfusion [7,8]. Both of these techniques share the advantage of being non-invasive and repeatable, similar to OCT. Although their use is currently limited in the clinical setting of PME, they should be kept in mind as they may be deployed as ancillary testing in atypical cases. It should be noted that, in subtle cases, PME may be asymptomatic and may only be diagnosed through imaging. The high specificity and sensitivity of OCT, complementedby its non-invasive nature, have made it the preferred diagnostic tool for PME in clinical practice, although FFA remains the gold standard [9].

Non-steroidal anti-inflammatory drugs (NSAIDs) remain the mainstay of treatment for PME due to their excellent efficacy. Since their topical application results in greater penetration of the aqueous and consequently the vitreous and retina, this is the preferred route for NSAID use [10]. Usually, topical NSAIDs are combined with topical steroids. Other treatment options include oral acetazolamide, triamcinolone delivered in the sub-tenon, the peribulbar or intravitreal route, as well as intravitreal dexamethasone implant. In refractory cases, intravitreal anti-vascular endothelial growth factor injections and tumor necrosis factor alpha inhibitors have rarely been employed [10].

Prostaglandin analogues (PGAs) are considered as the first-line treatment for glaucoma owing to their highly effective hypotensive action and good safety profile [11]. All currently used PGAs are related to $PGF_{2\alpha}$ and exert their action byrelaxing the ciliary muscles and elevating matrix metalloproteinase levels. These, in turn, they decompose collagen and increase the uveoscleral outflow of aqueous humor, thereby decreasing intraocular pressure (IOP) [12–15].

PGAs have been implied as risk factors for PME in complicated phacoemulsification; however, the evidence remains conflicting regarding uncomplicated cases [16]. An assortment of earlier and recent studies has reported an increased incidence of PME after perioperative PGA administration [17–20]. A few other recent publications seem to contradict these findings and advocate the safety of postoperative PGA use [21,22]. It is clear that a definitive study is still lacking. Most of the earlier studies used FFA to diagnose PME rather than the now more clinically relevant OCT. Even among those utilisingOCT, one did not exclude diabetes mellitus, a prominent confounding factor for PME, while Niyadurupola et al. had a follow-up of only 1 month [21,22].

Another point of clinical interest that few studies have examined is the efficacy of postoperative PGA use, particularly with concomitant topical NSAID administration. The meta-analysis by Lo et al. reportedincreased hypotensive action of PGAs when concomitantly administered with NSAIDs, but this has not been comprehensively evaluated in a postoperative setting [23].

Perhaps as a result of mixed findings in the available literature, PGA administration in the postoperative period of phacoemulsification surgery remains rather controversial in clinical practice. In a UK survey, 40.3% of ophthalmic surgeons elect to temporarily cease PGA administration following phacoemulsification, while in a similar Greek survey, 80% of surgeons reported the same [24,25]. In the absence of a clear consensus regarding the

postoperative use of PGAs, we aimed to further explore the safety of this practice with a focus onthe risk ofPME. We considered the 1-month follow-up of most studies inadequate to prove the safety of PGA use, as PME can also manifest in the late postoperative period. We therefore elected to follow our patients for 3 months. This also enabled us to examine the effects of re-initiating PGAs in the patient group that discontinued PGAs after their surgery. As a secondary outcome, we evaluated the efficacy of PGA administrationon IOP control in the early postoperative period after phacoemulsification surgery.

## 2. Materials and Methods

This prospective, randomised, comparative study was conducted at theGeneral Hospital Papageorgiou of Thessaloniki, Greece. It involvedpatients with ocular hypertension or early glaucoma who had been under PGA monotherapy and were scheduled for phacoemulsification surgery. To be eligible for this study, PGA treatment should have been initiated at least two months prior to the scheduled surgery and adequate IOP control should have been established. Patients with prior CME of any cause, diabetic macular edema, uveitis, uncontrolled glaucoma, glaucoma on combination therapy, age-related macular degeneration, or any other vision-limiting ocular disorder, as well as patients who had already undergone other ocular surgery on the study eye, were excluded.

The study was conducted according to the guidelines of the Declaration of Helsinki and approved by the Institutional Review Board of the General Hospital "Papageorgiou" (protocol code 0309, approved on the 28 January 2019).

After obtaining informed consent, eligible patients were randomly assigned to one of two groups using simple random number sampling on Microsoft Excel (Microsoft Corporation, Redmond, WA, USA). The first group of patients (PGA-on) continued PGA monotherapy for the full postoperative period, while the second group (PGA-off) ceased PGA use for 30 postoperative days and then reinitiated treatment. All patients were examined preoperatively and at one week, one month, and three months postoperatively. The major outcome of this study was the incidence of PME, diagnosed with spectral-domain OCT (Cirrus, Carl Zeiss Meditec, Dublin, CA, USA). Secondary outcomes included corrected distance visual acuity (CDVA), central macular thickness (CMT), average macular thickness (AMT), and IOP. CDVA was measured using early treatment diabetic retinopathy study (ETDRS) charts. CMT and AMT were calculatedutilising the macular cube OCT scan, according to the automated measurements of the macular map module. IOP was measured twice with Goldmann applanation tonometry and the mean was recorded. All measurements were performed by masked clinicians. All surgeries were performed by the same surgeon (EA) and on the same phacoemulsification machine (Centurion, Alcon, Switzerland). Patients with intraoperative complications, such as posterior capsular rupture or zonular dehiscence with or without vitreous prolapse, were excluded from the analysis. IOP > 28 mm Hg at any follow-up visit was considered an indication for additional rescue treatment and the patient was excluded from further analysis. The postoperative regime included a topical tobramycin/dexamethasone combination four times daily and topical nepafenac 0.1% three times daily for 1 month.

Numerical data were tested for normality of distribution with the Shapiro–Wilk test. Normally distributed values were analysed with a Student's *t*-test or a paired *t*-test for paired data. When the distribution was not normal, analysis was performed with the Wilcoxon Mann–Whitney rank sum test or the exact Wilcoxon signed-rank test for paired data. Categorical data were analysed with the Chi-square test, when its assumptions were met, or with Fisher's exact test. The sample size was calculated at 17 patients per group for a CMT difference of 10 μm and an IOP difference of 1 mm Hg between the groups ($\alpha = 0.05$, $\beta = 0.2$).

## 3. Results

Sixty-three eyes of sixty-threepatients scheduled for phacoemulsification surgery were randomised into two groups of either uninterrupted PGA use or temporary PGA cessation

for the first postoperative month. Seven eyes of seven patients (four from the PGA-on group and three from the PGA-off group) were lost to follow-up and were excluded from the analysis. Five eyes of five patients (four from the PGA-on group and one from the PGA-off group) developed IOP > 28 mm Hg during the follow-up period, requiring additional treatment, and were excluded from the analysis. The analysis included 22 eyes (22 patients) from the PGA-on group and 33 eyes (33 patients) from the PGA-off group. The mean age of our sample was 74.6 ± 6.6 years consisting of 32 women and 23 men ($p$ = 0.58). There were no significant differences in the age ($p$ = 0.45), diagnosis ($p$ = 0.33, Fisher's exact test), type of prostaglandin used ($p$ = 0.11), baseline CDVA ($p$ = 0.15, Wilcoxon Mann–Whitney rank sum test), and baseline IOP (PGA-on mean 16.1 ± 3.5 mm Hg; PGA-off mean 16.3 ± 3.8 mm Hg; $p$ = 0.85, Student's *t*-test) between the groups. We encountered a small but statistically significant difference in cumulative dissipated energy (CDE), which was higher in the PGA-on group (median CDE 13.45 in PGA-on; median CDE 10.4 in PGA-off; $p$ = 0.029). The baseline characteristics of the two groups can be seenin Table 1.

**Table 1.** Baseline characteristics.

| | PGA-On Group (*n* = 22) | PGA-Off Group (*n* = 33) | *p*-Value | 95% CI |
|---|---|---|---|---|
| **Eye** | | | | |
| Right no. | 13 | 16 | 0.442 * | |
| Left no. | 9 | 17 | | |
| **Gender** | | | | |
| Male no. | 10 | 13 | 0.58 ‡ | |
| Female no. | 12 | 20 | | |
| **Age** years, mean ± SD | 75.5 ± 8.1 | 73.9 ± 5.6 | 0.45 † | −6 to 3 |
| **Preoperative Treatment** | | | 0.11 ‡ | |
| Latanoprost no. | 14 | 10 | | |
| Tafluprost no. | 2 | 6 | | |
| Travoprost no. | 5 | 11 | | |
| Bimatoprost no. | 1 | 6 | | |
| **Diagnosis** | | | 0.33 ‡ | |
| Ocular hypertension no. | 6 | 4 | | |
| Primary open-angle glaucoma no. | 5 | 14 | | |
| Pseudoexfoliation syndrome no. | 9 | 8 | | |
| Normal tension glaucoma no. | 2 | 4 | | |
| Chronic closed-angle glaucoma no. | 0 | 3 | | |
| **CDE** median (IQR) | 13.45 (8.97) | 10.4 (6.66) | **0.029** † | −7.19 to −0.36 |
| **CDVA** logMAR, median (IQR)<br>Snellen, median (range) | 0.3 (0.16)<br>20/40 (20/132 to 20/28) | 0.4 (0.2)<br>20/50 (20/200 to 20/28) | 0.15 † | −0.06 to 0.18 |
| **IOP** mm Hg, mean ± SD | 16.1 ± 3.5 | 16.3 ± 3.8 | 0.85 ** | −1.8 to 2.2 |

* Chi-square test, ‡ Fisher's exact test, † exact Wilcoxon Mann–Whitney rank sum test, ** Student's *t*-test. Bold lettering is used to highlight statistically significant *p*-values. Abbreviations: CDE, cumulative dissipated energy; CDVA, corrected distance visual acuity; CI, confidence interval; IQR, interquartile range; logMAR, logarithm of the minimum angle of resolution; PGA, prostaglandin analogues; SD, standard deviation.

In this study, no eye developed PME within 3 months after phacoemulsification. Nonetheless, we noted a small but statistically significant increase in CMT and AMT in the 1st month and 3rd postoperative month. At the end of follow-up, the median CMT increase was 9 μm in the PGA-on group and 5 μm in the PGA-off group compared to baseline ($p$ < 0.001 in both groups, Wilcoxon signed-rank test). The median AMT increase was 10 μm in the PGA-on group and 11 μm in the PGA-off group compared to baseline ($p$ = 0.002 and $p$ < 0.001, respectively). The PGA-off group also presented this significant change in the 1st postoperative week ($p$ < 0.001). At each visit, no significant differences in CMT or AMT were noted between the two groups ($p$ > 0.05). Table 2 presents the AMT and CMT measurements at each visit.

**Table 2.** Central and average macular thickness duringall visits.

| | PGA-On Group (*n* = 22) | PGA-Off Group (*n* = 33) | *p*-Value | 95% CI |
|---|---|---|---|---|
| **CMT baseline** μm, mean ± SD | 250.5 ± 30.1 | 247.3 ± 23.6 | 0.66 * | −17.8 to 11.3 |
| **CMT 1st week** μm, mean ± SD | 253.4 ± 27.6 | 247.5 ± 23.7 | 0.40 * | −15 to 8 |
| *p*-value vs. baseline | 0.68 ‡ | 0.40 ‡ | | |
| 95% CI | −3.0 to 7.5 | −3 to 1.5 | | |
| **CMT 1st month** μm, mean ± SD | 259.7 ± 30.1 | 251.8 ± 25.2 | 0.32 * | −23.6 to 7.8 |
| *p*-value vs. baseline | **0.032 ⚹** | **0.018 ⚹** | | |
| 95% CI | 0.5 to 15.0 | 0.8 to 8.2 | | |
| **CMT 3rd month** mm Hg, mean ± SD | 265.1 ± 25.7 | 255.7 ± 25.4 | 0.18 * | −23.6 to 4.6 |
| *p*-value vs. baseline | **<0.001 ‡** | **<0.001 ‡** | | |
| 95% CI | 5.5 to 20.2 | 3.5 to 11.5 | | |
| **AMT baseline** μm, median, IQR | 266.0, 13.8 | 258.0, 24.0 | 0.55 † | −15 to 8 |
| **AMT 1st week** μm, median, IQR | 273, 22 | 270, 21 | 0.99 † | −10.3 to 10.1 |
| *p*-value vs. baseline | 0.11 ‡ | **<0.001 ‡** | | |
| 95% CI | −2.0 to 13.5 | 2.5 to 12.5 | | |
| **AMT 1st month** μm, median, IQR | 273.5, 21.5 | 271, 24 | 0.48 † | −14 to 7 |
| *p*-value vs. baseline | **0.001 ‡** | **<0.001 ‡** | | |
| 95% CI | 3.5 to 22.0 | 5.5 to 15.5 | | |
| **AMT 3rd month** μm, median, IQR | 277, 18.8 | 276, 22 | 0.71 † | −11.8 to 8.0 |
| *p*-value vs. baseline | **0.002 ‡** | **<0.001 ‡** | | |
| 95% CI | 5.5 to 20.0 | 7.5 to 19.5 | | |

* Student's *t*-test, ⚹ Paired *t*-test, † Wilcoxon Mann–Whitney rank sum test, ‡ Wilcoxon signed-rank test. Bold lettering is used to highlight statistically significant *p*-values. Abbreviations: AMT, average macular thickness; CI, confidence interval; CMT, central macular thickness; IQR, interquartile range; PGA, prostaglandin analogues; SD, standard deviation.

As expected, performing cataract surgery resulted in significantly increased CDVA compared to baseline for each group. At the end of follow-up, CDVA increased from 20/40 to 20/22 in the PGA-on group and from 20/50 to 20/25 in the PGA-off group ($p < 0.05$ during all postoperative visits). CDVA was similar between the two groups (1st month $p = 0.82$; 3rd month $p = 0.83$) with the exception of the first postoperative week, where we recorded significantly better CDVA in the PGA-off group (median value of 20/35 in PGA-on versus 20/28 in PGA-off; $p = 0.044$). We theorised that this transient difference was the result of significantly higher CDE in the PGA-on group. The analytical CDVA results are presented in Table 3.

**Table 3.** Corrected distance visual acuity during all visits.

| CDVA | PGA-On Group (*n* = 22) | PGA-Off Group (*n* = 33) | *p*-Value | 95% CI |
|---|---|---|---|---|
| **Baseline** logMAR, median (IQR) | 0.3 (0.16) | 0.4 (0.2) | 0.15 † | −0.06 to 0.18 |
| Snellen, median (range) | 20/40 (20/132 to 20/28) | 20/50 (20/200 to 20/28) | | |
| **1st week** logMAR, median (IQR) | 0.24 (0.24) | 0.14 (0.14) | **0.044 †** | −0.14 to 0 |
| Snellen, median (range) | 20/35 (20/50 to 20/20) | 20/28 (20/100 to 20/20) | | |
| *p*-value vs. baseline | **0.0075 ‡** | **<0.001 ‡** | | |
| 95% CI | −0.18 to −0.06 | −0.37 to −0.21 | | |
| **1st month** logMAR, median (IQR) | 0.12 (0.23) | 0.14 (0.1) | 0.82 † | −0.1 to 0.4 |
| Snellen, median (range) | 20/26 (20/50 to 20/20) | 20/28 (20/50 to 20/20) | | |
| *p*-value vs. baseline | **<0.001 ‡** | **<0.001 ‡** | | |
| 95% CI | −0.28 to −0.16 | −0.41 to −0.27 | | |
| **3rd month** logMAR, median (IQR) | 0.04 (0.13) | 0.1 (0.1) | 0.83 † | −0.04 to 0.6 |
| Snellen, median (range) | 20/22 (20/40 to 20/20) | 20/25 (20/40 to 20/20) | | |
| *p*-value vs. baseline | **<0.001 ‡** | **<0.001 ‡** | | |
| 95% CI | −0.33 to −0.19 | −0.41 to −0.28 | | |

† Wilcoxon Mann–Whitney rank sum test, ‡ Wilcoxon signed-rank test. Bold lettering is used to highlight statistically significant *p*-values. Abbreviations: CDVA, corrected distance visual acuity; CI, confidence interval; IQR, interquartile range; logMAR, logarithm of the minimum angle of resolution; PGA, prostaglandin analogues; SD, standard deviation.

The postoperative IOP values were similar between the two groups during the 1st week (PGA-on 17.4 mm Hg; PGA-off 17.5 mm Hg; $p = 0.93$), 1st month (PGA-on 15.5 mm Hg; PGA-off 16.0 mm Hg; $p = 0.88$), and 3rd month (PGA-on 13.2 mm Hg; PGA-off 13.0 mm Hg; $p = 0.72$) postoperatively. Compared to baseline, IOP was not significantly different during the 1st week (PGA-on $p = 0.31$; PGA-off $p = 0.15$) and during the 1st month (PGA-on $p = 0.94$; PGA-off $p = 0.67$). IOP was significantly lower during the 3rd month when the postoperative treatment had ceased and both groups had reinitiated PGAs (PGA-on IOP change $= -3$ mm Hg, $p < 0.001$; PGA-off IOP change $= -3.8$ mm Hg, $p < 0.001$). The IOP values during each visit are presented in Table 4.

**Table 4.** Intraocular pressure during all visits.

| Intraocular Pressure | PGA-On Group (*n* = 22) | PGA-Off Group (*n* = 33) | *p*-Value | 95% CI |
|---|---|---|---|---|
| **Baseline** mm Hg, mean ± SD | 16.1 ± 3.5 | 16.3 ± 3.8 | 0.85 * | −1.8 to 2.2 |
| **1st week** mm Hg, mean ± SD | 17.4 ± 6.0 | 17.5 ± 4.1 | 0.93 * | −2.6 to 2.9 |
| *p*-value vs. baseline | 0.31 ‡ | 0.15 ‡ | | |
| 95% CI | −1.2 to 3.7 | −0.45 to 2.8 | | |
| **1st month** mm Hg, median(IQR) | 15.5 (4.8) | 16.0 (4.0) | 0.88 † | −3 to 3 |
| *p*-value vs. baseline | 0.94 ‡ | 0.67 ‡ | | |
| 95% CI | −2.6 to 2.4 | −2.3 to 1.5 | | |
| **3rd month** mm Hg, mean ± SD | 13.2 ± 2.4 | 13.0 ± 2.8 | 0.72 * | −1.7 to 1.2 |
| *p*-value vs. baseline | **<0.001 ‡** | **<0.001 ‡** | | |
| 95% CI | −4.2 to −1.6 | −6.0 to −1.5 | | |

\* Student's *t*-test, ‡ Paired *t*-test, † Wilcoxon Mann–Whitney rank sum test. Bold lettering is used to highlight statistically significant *p*-values. Abbreviations: CI, confidence interval; IQR, interquartile range; PGA, prostaglandin analogues; SD, standard deviation.

## 4. Discussion

In this prospective randomised study, we evaluated the safety of topical PGA continuation following uncomplicated phacoemulsification surgery. All surgeries were performed by the same surgeon in a sample of Caucasians of Greek origin with a 3-month follow-up period. Our results support the safety of PGA continuation after phacoemulsification, with no events of PME in either group.

In this sample, no incidence of PME was recorded on macular OCT, and CMT andAMT were similar in both groups during each visit. It should be noted that the standard postoperative medication included topical NSAIDs for 1 month, which can treat clinical and subclinical PME [19]. In both groups, we recorded a small but statistically significant increase in CMT compared to baseline until the end of follow-up. This small change was described as a physiological increase in retinal thickness following phacoemulsification and was attributed to either subclinical inflammation or changes in the measurement technique, predominantly in signal strength index [26,27]. CDVA was also similar between the two groups with the exception of the first postoperative week where we noted slightly better CDVA in the PGA-off group. Perhaps this could be attributed to the higher CDE in the PGA-on group.

Our findings are in accordance with most relevant prospective studies. The RCT by Niyadurupola et al. showed identical incidences of PME in patients who either continued or discontinued PGAs following uneventful phacoemulsification [21]. Fakhraie et al. focused on latanoprost administration and reported no events of PME, similar to the present study, even without NSAIDs or the exclusion of patients with diabetes mellitus. Furthermore, they recorded a transient increase in CMT post-phacoemulsification regardless of PGA use, similar to the present study [22]. The RCT by Park et al. focused specifically on changes in macular thickness after phacoemulsification and included a control group of patients without glaucoma. Their results with only one case of PME in the control group also support the continued use of GPAs with concomitant topical NSAID administration [28]. Nonetheless, a number of prospective studies have reported the opposite, with an increased incidence of PME after perioperative PGA administration [18–20]. Miyake et al. showed

an increased incidence of angiographic PME with latanoprost alone, but this was easily reversed when concurrently administering topical diclofenac [19]. Lee et al. performed multiple regression analyses on their non-randomised prospective study and reported a correlation of PGA use withincreased incidence of PME. However, diabetes mellitus was not excluded and NSAIDs were prescribed only uponPME diagnosis [20]. A large, nested case–control study with more than 5000 eyes demonstrated an increased risk ratio (RR) for PME in PGA users after excluding diabetes mellitus. Among the retrospective studies, Wendel et al. analysed each PGA agent in isolation showing a significant association betweentravoprost use (RR 3.16) and bimatoprost use (RR 2.06), but not latanoprost (RR 1.55), with the development of PME within a year from phacoemulsification surgery [17]. The largest database study to date was conducted by Chu et al. and included 81,194 eyes of patients with or without diabetes mellitus. In the subgroup analysis where diabetics were excluded, PGA was not a significant risk factor for PME development [4].

We demonstrated similar IOP values in both groups and during all visits regardless of continuing or ceasing topical PGAs. This was a rather surprising finding considering the meta-analysis by Lo et al. who suggested an enhanced IOP-lowering effect of combined NSAID and PGA administration. It should be noted that the meta-analysis did not address their perioperative use [23]. Fakhraie et al. reported similar IOP values between groups regardless of latanoprost continuation until the 3rd postoperative month [22]. In contrast, Miyake et al. and Niyadurupola et al. reported lower IOP in the PGA-continuation groups, although the former did not routinely administer NSAIDs [19,21]. In this study, we noted a non-significant decrease in IOP in both groups compared to baseline after the first postoperative month. We hypothesised that the transient hypotensive effect of phacoemulsification surgery allowed the PGA-off group to maintain normal IOP [29]. After the first month, topical NSAIDs were discontinued and the PGA-off group reinitiated PGA treatment. At the end of follow-up, IOP was significantly lower compared to baseline in both groups, which we attributed to the IOP-lowering effect of phacoemulsification, while maintaining no difference between groups.

The evidence on the safety of PGA continuation is limited in the available literature. This has resulted in many clinicians discontinuing PGAs after phacoemulsification surgery. The present prospective study features a relatively longer follow-up compared to similar studies, while also covering the re-initiation of treatment in the group that discontinued PGAs. Furthermore, in accordance with recent studies, we used OCT to detect PME based on its exceptional sensitivity and wider clinical application compared to fluorescein angiography. Naturally, the present study has some weaknesses; the sample is relatively small and comprises patients under various PGA regimens and diagnoses, including OHT. Despite randomization, the PGA-on group had a statistically higher CDE, but this would have been expected to increase the incidence of PME [30]. Therefore, this may further reinforce our results. Another limitation of this study is the routine prescription of topical NSAIDs which display a protective effect against PME and may act as a confounding factor. Nonetheless, post-operative NSAIDs are considered as the standard of care in Greece, and our goal was to investigate the effects of continuing PGAs in a setting relevant to our clinical practice.

The present study supports the safety of continued PGA use in uncomplicated cataract surgery with concurrent topical NSAID use. The option to continue PGA use in the early postoperative period appears safe regarding the incidence of PME and the final decision could be left to the discretion of the ophthalmic surgeon. Larger studies may be required to solidify the safety of PGAs in the early postoperative period.

**Author Contributions:** Conceptualization: E.A. and A.T.; methodology: E.A. and A.T.; validation: N.Z.; formal analysis: S.K. and A.T.; investigation: A.M. and M.D.; resources: N.Z. data curation: A.M., M.D. and A.T.; writing—original draft preparation: S.K.; Writing—reviewing and editing: E.A., S.K. and A.T.; visualization: E.A., N.Z. and A.T.; supervision: E.A. and N.Z.; project administration: E.A. and N.Z.; funding acquisition: none. All authors have read and agreed to the published version of the manuscript.

**Funding:** No funding or other kinds of support were obtained for the realization of this study.

**Institutional Review Board Statement:** The study was conducted according to the guidelines of the Declaration of Helsinki and approved by the Institutional Review Board of General Hospital "Papageorgiou" (protocol code 0309, approved on the 28 January 2019).

**Informed Consent Statement:** Informed consent was obtained from all subjects involved in the study.

**Data Availability Statement:** The data presented in this study are available on request from the corresponding author.

**Conflicts of Interest:** The authors declare no conflict of interest.

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
