# Peer review of "Safety and Efficacy of Prostaglandin Analogues in the Immediate Postoperative Period after Uneventful Phacoemulsification"

_2411-5150, 2023_

Round 1

Reviewer 1 Report

The sample should be expanded. This paper is not particularly original nonetheless is interesting .

Author Response

  • The sample should be expanded.

Thank you for your comment and your suggestion. We agree that a bigger sample would reinforce our findings, and we refer this in the limitations of this study (line 283): “Naturally, the present study has some weaknesses; the sample is relatively small”

Nonetheless we should note that the statistical power was adequate to detect a 10μm difference in CMT and 1mmHg difference in IOP between the two groups, as stated in line 147: “The sample size was calculated at 17 patients per group for a CMT difference of 10 μm and an IOP difference of 1 mmHg between groups (α=0.05, β=0.2).”

  • This paper is not particularly original nonetheless is interesting.

Thank you for your comment. Although this topic has been researched before, few studies reported IOP or followed the patients prospectively for more than 1 month. We believe that it is important to add these two points in the discussion, especially when advocating for PGA continuation.

Reviewer 2 Report

Good study, conducted according to the principles.

Reasons for excluding 7 patients from study group?

Author Response

  • Good study, conducted according to the principles.

    Reasons for excluding 7 patients from study group?

Thank you for your kind comment. We excluded 7 patients from analysis because they did not return for the follow-up visits, as well as 5 patients for requiring further antiglaucoma medication due to developing IOP>28mmHg. This is mentioned in line 154: “Seven eyes of 7 patients (4 from the PGA-on group and 3 from the PGA-off group) were lost to follow-up and were excluded from analysis. Five eyes of 5 patients (4 from the PGA-on group and 1 from the PGA-off group) developed IOP >28 mmHg during the follow-up period, requiring additional treatment, and were excluded from analysis”. We believe that excluding these outliers gives a more robust result regarding our principal research question.

Reviewer 3 Report

This interesting manuscript has two major limitations:

a. 55 eyes (22 + 33) are a small sample, too small to draw any major conclusion.

b. the other flaw is represented by the postsurgery therapeutic regime. According to the authors, it included topical NSAID (nepafenac 0.1% three times daily for 1 month), as it is commonly made. A potential PGAs pro-PME effect cannot be determined accurately if a NSAID is contemporarily administered.

It may be guessed that if the study protocol had included a group of eyes with postsurgery therapy without NSAID, the Hospital Institutional Review Board would not have approved the study.

The clinical meaning of the study would benefit a lot by adding a third group of PGA-on eyes not receiving topical NSAID and observing what happens in terms of PME.

On the other side, it may be suggested that topical NSAID use is “protective” since after cataract surgery it probably allows to continue PGAs in glaucomatous eyes that cannot discontinuate therapy

Author Response

  • This interesting manuscript has two major limitations:
    a. 55 eyes (22 + 33) are a small sample, too small to draw any major conclusion.

Thank you for your comments. We agree that the sample is relatively small, which we outline as a weakness of this study in line 281: “Naturally, the present study has some weaknesses; the sample is relatively small”. Nonetheless, it should be noted that the statistical power was adequate for the detection of IOP differences of 1mmHg and CMT difference of 10μm between the two groups. Indeed, a major conclusion regarding CME incidence after continued PGA administration would perhaps require a study with an epidemiological focus and far more patients.

  • b. The other flaw is represented by the postsurgery therapeutic regime. According to the authors, it included topical NSAID (nepafenac 0.1% three times daily for 1 month), as it is commonly made. A potential PGAs pro-PME effect cannot be determined accurately if a NSAID is contemporarily administered. It may be guessed that if the study protocol had included a group of eyes with postsurgery therapy without NSAID, the Hospital Institutional Review Board would not have approved the study.
    The clinical meaning of the study would benefit a lot by adding a third group of PGA-on eyes not receiving topical NSAID and observing what happens in terms of PME.
    On the other side, it may be suggested that topical NSAID use is “protective” since after cataract surgery it probably allows to continue PGAs in glaucomatous eyes that cannot discontinuate therapy,

Thank you for your insight. We agree that NSAIDs may act as confounding factors when investigating the direct effect of PGAs. Nonetheless, NSAIDs are routinely prescribed after phacoemulsification in Greece and we wanted our study to be as close to standard clinical practice as possible.
As a follow-up to your comment we have added this point to the limitations of this study in the discussion section, line 287, as follows: “Another limitation of this study is the routine prescription of topical NSAIDs which display a protective effect against PME and may act as a confounding factor. Nonetheless, post-operative NSAIDs are considered standard of care in Greece and our goal was to investigate the effects of continuing PGAs in a setting relevant to our clinical practice”.
We also updated the abstract to include the concomitant NSAID use: “Topical non-steroidal anti-inflammatory drugs (NSAIDs) were routinely administered in all patients during the first postoperative month.”, “In conclusion, PGA administration with concomitant topical NSAIDs appears a safe practice in the early postoperative period of uncomplicated phacoemulsification”.

Round 2

Reviewer 3 Report

Accept as it stands